# Transcriptomic Signature of the Simulated Microgravity Response in *Caenorhabditis elegans* and Comparison to Spaceflight Experiments

**DOI:** 10.3390/cells12020270

**Published:** 2023-01-10

**Authors:** İrem Çelen, Aroshan Jayasinghe, Jung H. Doh, Chandran R. Sabanayagam

**Affiliations:** 1Center for Bioinformatics and Computational Biology, University of Delaware, Newark, DE 19711, USA; 2Delaware Biotechnology Institute, University of Delaware, Newark, DE 19711, USA

**Keywords:** ceramide, sphinogolipid signaling, longevity, transcriptome, space, intergenerational, microgravity

## Abstract

Given the growing interest in human exploration of space, it is crucial to identify the effects of space conditions on biological processes. Here, we analyze the transcriptomic response of *Caenorhabditis elegans* to simulated microgravity and observe the maintained transcriptomic response after returning to ground conditions for four, eight, and twelve days. We show that 75% of the simulated microgravity-induced changes on gene expression persist after returning to ground conditions for four days while most of these changes are reverted after twelve days. Our results from integrative RNA-seq and mass spectrometry analyses suggest that simulated microgravity affects longevity-regulating insulin/IGF-1 and sphingolipid signaling pathways. Finally, we identified 118 genes that are commonly differentially expressed in simulated microgravity- and space-exposed worms. Overall, this work provides insight into the effect of microgravity on biological systems during and after exposure.

## 1. Introduction

Significant efforts are being made for the human exploration of space. It is known that space conditions can negatively affect biological processes by causing bone loss, muscle atrophy, and immune system impairment, to name a few examples (reviewed in [1,2]). However, the underlying molecular mechanisms that contribute to these adverse effects is largely unknown.

Microgravity is one of the major stress factors causing detrimental health effects on humans (reviewed in [3]). Simulation of microgravity is a cost-effective way to study the impact of microgravity on biological systems. For this purpose, various platforms including drop towers, parabolic flights and space flights are available. Because of their advantages to enable long term simulations in an effective and economical manner, clinostats are widely preferred to study gravitational response in biological systems, mainly plants, for decades [4,5,6,7,8]. Clinostats rotate a sample around a horizontal axis, thereby exposing the sample to a rotating gravitational vector. This can reduce and even possibly remove gravitational bias in the development of an organism [6,9,10,11,12,13].

*C. elegans* is an exceptional model organism for space biology studies [14] with its completely documented cell lineage, high reproduction rate, short lifespan, and high similarity to the human genome (reviewed in [15]). The usage of this organism in previous space missions (such as International *C. elegans* Experiment in Space (ICE-First), *C. elegans* RNAi space experiment (CERISE), Shenzhou-8, and Commercial Generic Bioprocessing Apparatus Science Insert-01 (CSI-01)) have provided valuable insights into the biological response of *C. elegans* to space conditions [16,17,18,19,20,21,22,23,24]. However, these studies generally have utilized different strains and liquid cultures instead of commonly used OP50-seeded agar plates to reduce the impact of surface tension [25]. We previously have shown that the usage of different liquid cultures and strains cause highly distinct intergenerational transcriptomic and phenotypical responses [26]. Similarly, a recent study reported dramatical intergenerational changes in the physiology of liquid-grown worms [27]. Thus, complementary to the spaceflight studies, separate investigations can provide insight into the impact of individual space conditions on biological processes.

This study investigates the effect of simulated microgravity and the sustained impacts after return to ground conditions on *C. elegans* transcriptome. Through the RNA-sequencing (RNA-seq) and mass spectrometry analyses, we reveal the downregulation of the sphingolipid signaling pathway under simulated microgravity. In addition, we identify a putative microgravity-responsive transcriptomic signature by comparing our results with previous studies.

## 2. Materials and Methods

### 2.1. C. elegans Strain and Growth Conditions

Wild-type N2 strain was obtained from the Caenorhabditis Genetics Center (CGC). The worms were grown at 21 °C. Stocks of *C. elegans* were acclimated to CeHR medium for three weeks prior to microgravity experiments. The ground control animals were maintained in CeHR medium in 20 mL scintillation vials as described [26]. Approximately 100,000 live worms (P_0_ generation) were cleaned by sucrose floatation, their embryos (F_1_ generation) were harvested by bleaching and placed in ~40 mL of CeHR culture in sterile VueLife culture bags (“AC” Series 32-C, Saint-Gobain). Appendix A show some of the worms in the VueLife culture bags. Culture bags were mounted in the clinostat and rotated at 1 rad/s (~10 RPM) for four days, which is the measured time to reach L4/adulthood [28]. The “F_2_” and subsequent generations were approximated by using four-day intervals. Prior to RNA extraction, only live worms are retrieved using the sucrose floatation method. We note that “F2” generation and beyond will not maintain synchronicity, and all will contain a mixture of generations, but recovering only the live animals enriches the RNA pool for the expected (majority) generation as each adult will produce ~30 offspring. The reason for the usage of mixed stage worms were the difficulties experienced (e.g., contamination of the liquid medium) during the synchronizing of the animals.

### 2.2. RNA Isolation, Illumina Sequencing

Wild-type N2 strain was obtained from the Caenorhabditis Genetics Center (University of Minnesota, Twin Cities, MN, USA). The worms were grown at 21 °C. Stocks of *C. elegans* were acclimated to CeHR medium for three weeks prior to microgravity experiments. Approximately 100,000 live worms (P0 generation) were cleaned by sucrose floatation, their embryos (F1 generation) were harvested by bleaching and placed in ~40 mL of CeHR medium in sterile VueLife culture bags (“AC” Series 32-C, Saint-Gobain, La Défense Cedex, France). Four culture bags were mounted in a lab-built clinostat inside a laminar flow biosafety hood, and one was kept aside as a ground control. The clinostat rotated the culture bags at 1 rad/s (~10 RPM) for four days, which is the measured time to reach L4/adulthood at 21 °C in CeHR medium [28]. Worms from one bag were immediately harvested to represent simulated microgravity conditions. The remaining three culture bags were processed in four-day intervals to represent the return to ground conditions. We note that F2 generation and beyond will not maintain synchronicity, and all will contain a mixture of generations, but recovering only the live animals enriches the RNA pool for the expected (majority) generation as each adult will produce ~30 offspring in CeHR culture. The experimental scheme is limited to using mixed stage worms because of the technical difficulties associated with manually picking the desired age (swimming) worms from liquid cultures. Live worms were selected for harvesting by sucrose floatation and total RNA was isolated with TRIzol treatment and recovered by alcohol precipitation. Total RNA was further purified by PureLink ™ RNA Mini Kit (Life Technologies, Carlsbad, CA, USA). Two micrograms of total RNA were used for library preparations using the TruSeq Stranded mRNA LT Sample Prep Kit (Illumina). Libraries were sequenced on an Illumina HiSeq 2500 instrument set to the rapid run mode using single-end, 1 × 51 cycle sequencing reads as per manufacturer’s instructions. RNA-seq was performed for each condition with three replicates.

### 2.3. Gene Expression Analysis

Quality control on the RNA-seq data was done with FASTQC (version 0.11.2) [29]. All the reads were in the “very good” quality range. The DEGs were identified by using two separate approaches. First, the Tuxedo pipeline [30] were used with default parameters. Second, the DESeq2 software (version 1.16.1) [31] was utilized after quantifying the expression of transcripts with Salmon (version 0.8.2) [32]. The reference genome (WBCel235) along with the annotation file were retrieved from Ensembl [33]. We considered genes as differentially expressed if FDR adjusted *p*-values < 0.05, and log2 fold change > 2 unless otherwise stated. The results from the Tuxedo pipeline were used throughout the paper as explained in the results and the results from DESeq2 were reported in the Appendix A (Appendix A; Appendix A). We discarded miRNA, piRNA, and rRNA molecules from the analysis. The ncRNA molecules (WS250) and unconfirmed genes were obtained from WormBase [34]. Dendrogram was generated with Ward’s clustering criterion by using R software (version 3.5.1).

### 2.4. Functional Analysis of the Genes

Gene ontology (GO) enrichment for biological processes and protein domains were assessed with Database for Annotation, Visualization and Integrated Discovery (DAVID) (v6.8) [35]. The GO network for the common DEGs between our simulated microgravity experiment and four-day CERISE was created in Cytoscape (version 3.4.0) [36] by using the BINGO plugin [37]. Pathway analysis was done using KEGG Mapper (https://www.genome.jp/kegg/tool/map_pathway1.html (accessed on 1 January 2019)). Tissue enrichment analysis was performed by using WormBase Gene Set Enrichment Analysis tool [38], and FDR adj. *p*-value  <  0.05 considered significant and top ten tissues were reported.

### 2.5. Comparison of DEGs to the CERISE Four Days

NASA’s GeneLab (https://genelab.nasa.gov/ (accessed on 1 January 2019)) platform was used to acquire the Gene Expression Omnibus (GEO) accession number for the previous studies on microgravity related gene expressions in *C. elegans*. We identified four such studies with the following GEO accession numbers: GSE71771 (Expression Data from International *C. elegans* Experiment 1st), GSE71770 (four-day CERISE), GSE27338 (eight-day CERISE), and GSE32949 by UKM (PRJNA146465) from the query made on 30 June 2016 with “*C. elegans*” and “microgravity” terms on GeneLab. We used data from only four-day CERISE as the duration of the study is the same with ours. We analyzed the publicly available microarray datasets with GEO2R (https://www.ncbi.nlm.nih.gov/geo/geo2r/ (accessed on 1 January 2019)) for gene expression under microgravity at the ISS versus 1G control in four day CERISE. The genes with logFC > 2 and Benjamini and Hochberg corrected *p*-value < 0.05 were considered as differentially expressed.

STRING database (version 10.5) [39] was used to determine whether the common DEGs from our simulated microgravity experiment and four-day CERISE show enrichment for protein–protein interactions (PPI). We only considered the interactions only from experiments, gene-fusion, databases, and co-expression and selected “high-confidence (0.7)” results. We excluded the interactions based on text-mining, fusion, neighborhood, and co-occurrence as we were only interested in PPI with higher experimental and/or computational confidence. PPI enrichment showed significance at *p*-value < 1.0 × 10^−16^ with 58 interacting nodes and 155 edges. The PPI network was visualized with Cytoscape (version 3.4.0) [36]. Edge width was determined based on the “combined score” for the interactions obtained from the STRING database. We identified the human orthologs of the common DEGs in our experiment and four-day CERISE with OrthoList 2 with default settings [40].

### 2.6. Mass Spectrometry for Ceramides

The worms were chopped manually with razor blades on ice-cold glass. Three replicates from mixed stage worms grown in ground control and simulated microgravity conditions were used simultaneously. Each sample was sent to Avanti Polar Lipids Analytical Services as a frozen extract in glass tubes for liquid chromatography with tandem mass spectrometry (LC/MS/MS) experiment for detection of Ceramide levels. Each sample was extracted by modified Bligh and Dyer extraction [41]. The samples were dried and resuspended in 50:50 Chloroform:Methanol and diluted prior to analysis. The resolved sample was used for analyses and stored at −20 °C until assayed. Samples were diluted as needed for analysis with internal standards for Ceramides for quantization by injection on LC/MS/MS. The individual molecular species for each sphingolipid group were measured by reversed phase liquid chromatography tandem mass spectrometry methods which separate the compounds by multiple reaction monitoring *m*/*z* to fragments and retention time. Quantification was performed by ratio of analyte to internal standard response multiplied by ISTD concentration and *m*/*z* response correction factor. The results are expressed as ng/rnl. The consistency among the replicates were examined with Pearson correlation. All but one replicates in both the experiments showed high correlation (R^2^ > 0.9). The ground control replicate which did not show a strong correlation (around 60%) with the other replicates was excluded from the analysis. The data were analyzed with *t*-test (two-tailed), and *p*-value < 0.05 was considered significant. Equal variances were confirmed with Levene’s test at α = 0.05 for all the ceramides tested.

### 2.7. Microgravity Simulation with Clinorotation

We used a laboratory-built clinostat with an integrated microscope to observe the motion of small objects including microspheres, yeast cells, and *C. elegans* embryos during clinorotation. The trajectory of a spherical particle in a rotating vessel has been described previously in literature [5,11,42,43]. Dedolph and Dipert (1971) separated the motion of a spherical particle into two parts: a circular motion under the influence of gravity, the radius of which is inversely proportional to the angular velocity (*ω*, rad/s), and a radial motion due to centrifugal forces, the velocity of which is proportional to *ω*^2^ [42]. The equation of motion that describes the orbital trajectories is given by:$$m\frac{d^{2}q}{dx^{2}}+\left(2i\omega m+\lambda \right)\frac{dq}{dt}-\omega^{2}\left(m-m_{w}\right)q=-ig\left(m-m_{w}\right)e^{-i\omega t}.$$

In the above equation, *λ* is the viscous drag coefficient, *m* is the mass of the object, *m_w_* is the mass of the displaced water, *t* is time, *g* is the acceleration due to gravity, and *q*(*x*,*y*,*z*,*t*) is the rotating reference frame defined as:$$q\equiv x_{rot}+iy_{rot}=ze^{-i\omega t}.$$

As a test of the performance of the clinostat we recorded the motion of fluorescent melamine-formaldehyde microspheres (5.6 µm diameter, 1.51 g/cm^3^ density; Corpuscular Inc., Cold Harbor, NY, USA) suspended in a 3% bovine serum albumin (BSA) solution (Appendix A). This solution was injected into an 8 mm diameter, 100-µm deep, coverglass topped chamber mounted on the clinostat microscope. Some of the melamine microspheres adhere to coverglass of the chamber, conveniently providing fiducial markers which we were able to use to compensate for the mechanical noise introduced into the micrographs by the rotation of the clinostat. We measured the terminal velocity of the suspended microspheres to be 3.2 ± 0.2 µm/s (mean ± SEM, *n* = 8). Appendix A shows a plot of the measured radius of circular motion of suspended microspheres as a function of inverse angular velocity (black dots), and the calculated radius [5] using the measured mean terminal velocity (blue line). The shaded area encompasses calculated radii for terminal velocities within one standard deviation of the mean.

## 3. Results

To dissect the biological processes affected under simulated microgravity and sustained after the exposure, we first cultured *C. elegans* in CeHR for three weeks on ground control condition and allowed the worms to acclimate to the liquid culture. Then, we exposed the worms to clinostat-simulated microgravity for four days (Figure 1a; Appendix A) and observed the maintained impacts at four, eight, and twelve days after placing the worms back to ground conditions (Figure 1b). RNA-seq was performed for each condition with three replicates (Appendix A). Because RNA-seq can detect low abundance transcripts and achieve less noise in the data, unlike previous space biology studies on *C. elegans*, we preferred RNA-seq over microarray [44].

We generated a dendrogram for the transcriptomic profile in each condition tested and found significant differences during and after exposure to simulate microgravity compared to the ground control conditions (Figure 1c). Thus, this result suggests that exposure to simulated gravity induces highly distinct gene expression patterns which are maintained even after 12 days return to ground conditions (approximately three generations of *C. elegans* in axenic medium [28]).

### 3.1. Simulated Microgravity Triggers Differential Expression of Hundreds of Genes

To identify the genes with the most distinctive expression levels after the exposure, we determined the differentially expressed genes (DEGs) in the simulated microgravity-exposed and returned worms against the ground control. The genes with over two-fold log2 expression difference with the FDR-adjusted *p*-value ≤ 0.05 were considered as DEGs. Hundreds of genes demonstrated differential expression during exposure to simulated microgravity and up to eight days after return to ground conditions (Figure 1d). Twelve days after the return, the gene expression levels started to resemble the ones in the ground control with only 91 upregulated and 13 downregulated genes.

The spatial expression of the DEGs is of great importance to identify the potentially affected tissues. Thus, we performed tissue enrichment analysis for the DEGs in simulated microgravity (Figure 2a). The downregulated genes were overrepresented in neuronal and epithelial tissues, and intestine while the upregulated genes were enriched in the reproductive system-related tissues.

Next, we conducted pathway enrichment analysis for the DEGs in simulated microgravity and return to ground conditions to identify the altered pathways and whether they remained altered after return to ground conditions (Figure 2b). Our results suggested an upregulation of dorso–ventral axis formation and downregulation of lysosome during the exposure, and these expression patterns were maintained for four and eight days after the return, respectively.

### 3.2. Simulated Microgravity-Induced Gene Expression Differences Are Highly Maintained for Eight Days after Return to Ground Conditions

We identified the number of simulated microgravity-induced DEGs that preserved their expression patterns after the return to ground control (Figure 2c). The majority of the DEGs (approximately 75%) from the exposed animals maintained their expression patterns after the return for four days. The shared number of DEGs decreased drastically at 12 days after the return to ground conditions: 16% for commonly upregulated and <1% downregulated genes with the exposed animals.

To elucidate the genes showing altered expression under simulated microgravity and maintaining these expression patterns after return to ground conditions for short (four days) and long-term (12 days), we categorized the DEGs in Venn diagrams (Figure 2d,e; Appendix A). The genes solely upregulated in the simulated microgravity exposed animals did not exhibit enrichment for any gene ontology (GO) term. The genes upregulated during the exposure and four days after the return, however, showed an overrepresentation for reproduction-related processes (Figure 2d). Chitin metabolic process-related genes were induced at four days after return to ground conditions, and the expression profiles were conserved for eight days after the return.

We found that the downregulated genes were enriched for the biological processes which are affected in space conditions. In particular, genes functioning in body morphology, collagen and cuticulin-based cuticle development, defense response, and locomotion were downregulated under simulated microgravity and up to eight days after return to ground conditions (Figure 2e). In simulated microgravity and spaceflight studies, both small movement defects [18] and no difference in locomotory behavior [19,45] have been noted. The reason behind these discrepancies is unclear, but our results indicated differences in the expression of the locomotion genes in close to weightless environment. Similarly, neuropeptide signaling pathway genes (*flp-22*, *flp-8*, *flp-26*, *nlp-10*, *nlp-12*, *nlp-17*, *nlp-20, nlp-24*, *nlp-25*, *nlp-26*, *nlp-39, nlp-33*, *nlp-28*, *nlp-29*, *nlp-30,* and *flp-24*) were downregulated during the exposure, and this expression profile was lost after return to ground conditions (Figure 2e). These neuropeptide signaling pathway genes also exhibit enrichment for movement variant phenotypes (Appendix A) indicating altered locomotion in response to simulated microgravity.

During spaceflight, the neuromuscular system and collagen are negatively affected [18]. In agreement with these findings, collagen genes were downregulated in our experiment during the exposure, and the downregulation was sustained for eight days after return to ground conditions. Previously, Higashibata et al. (2006) reported decreased expression for the myogenic transcription factors and myosin heavy chains in space-flown worms [18]. We did not observe differential expression for these muscle-related genes. Since their reported flight to ground gene expression ratio for the downregulated genes was less than one, and we only consider the genes with the log2 ratio greater than two as DEG, we did not expect to observe those genes in our group of DEGs. To test whether other muscle-related genes are differentially expressed under simulated microgravity, we compared our DEGs to 287 genes reported in WormBase for involvement in muscle system morphology variant or any of its transitive descendant terms via RNAi or variation. Only T14A8.2, *col-103,* and *sqt-3* among the 287 genes showed downregulation under simulated microgravity (hypergeometric test, *p*-value = 0.99). Thus, our results did not suggest a significant change in the expression of the genes known to function in muscle morphology.

In our previous work, we revealed that the expression of many ncRNA molecules is triggered in response to environmental changes [26]. To identify whether a similar pattern occurs under simulated microgravity, we determined the expressed ncRNA molecules in ground control, simulated microgravity, and return. Since the housekeeping gene *pmp-3* presents consistent expression patterns both in our study and previous studies [26], we considered its minimum expression level (FPKM = 24.7) in our experiment as the cutoff for the expression of ncRNA molecules. The number of expressed ncRNA molecules slightly decreased in response to simulated microgravity (Appendix A). Interestingly, this number increased on the eighth day of return. We found that the expression of 126 ncRNA molecules is induced during simulated microgravity exposure and for 12 days after the return (Appendix A) while the expression of 16 ncRNA molecules is lost during and after simulated microgravity (Appendix A). Among the classified ncRNA molecules, mostly small nucleolar RNA (snoRNA) and long noncoding RNA (lincRNA) molecules were induced while other sets of snoRNA and antisense RNA (asRNA) molecules lost expression in simulated microgravity. For instance, asRNA molecules *anr-33*, K12G11.14, and ZK822.8 are induced whereas *anr-2*, *anr-9*, and Y49A3A.6 are silenced during and 12 days after the exposure.

We next sought to identify the putative transcriptional regulators (i.e., transcription factors) of the simulated microgravity-induced genes and whether these regulators maintain their impact after return to ground conditions. That is, we determined the putative transcription factor (TF) genes [46] that are upregulated under simulated microgravity and after the return to ground conditions. Our results have revealed that 20 TF genes are upregulated during the exposure and 90%, 75%, and 15% of these genes maintained their upregulation after the return for four, eight, and twelve days, respectively (Appendix A). These TFs play a role in a variety of mechanisms such as double strain break repair or sex determination. For example, *tbx-43* is upregulated during the exposure and for at least 12 days after the return to ground. The best human BLASTP-match of *tbx-43* in WormBase, TBR, functions in developmental processes and is required for normal brain development (E-value = 3 *×* 10^−33^; identity = 68.5%). However, many of the upregulated TFs do not have a known function. Hence, our list of simulated microgravity-induced TF gene expressions can be a rich source for the discovery of the microgravity-related transcriptional regulators.

### 3.3. Longevity Regulating Pathways Are Affected under Simulated Microgravity

Spaceflight affects the mechanisms involved in delayed aging in worms. For instance, Honda et al., found that age-dependent increase in 35-glutamine repeat aggregation is suppressed and seven longevity-controlling genes are differentially expressed in spaceflight [20,24]. To examine whether simulated microgravity induces such changes, we mapped the DEGs (simulated microgravity versus ground control) to longevity regulating pathways in the Kyoto Encyclopedia of Genes and Genomes (KEGG). We used a less stringent criterion for differential expression by including the genes with log_2_(FPKM) > 1.5 in this analysis. We determined 11 DEGs in the longevity regulating pathway genes most of which are involved in the insulin/insulin-like growth factor signaling pathway (insulin/IGF-1) (Figure 3). Interestingly, the transcription factor DAF-16 gene did not exhibit differential expression unlike its targets *sod-3*, *lips-17*, and *ctl-1*. Because the translocation of DAF-16 into the nucleus activates or represses target genes functioning in longevity, metabolism, and stress response, this finding was unexpected [47,48,49,50]. The nuclear localization of DAF-16 is antagonized by transcription factor PQM-1 [51], and thus an upregulation in *pqm-1* may indicate inhibition of DAF-16 translocation. Our data, however, did not present a differential expression for *pqm-1* and thus it does not suggest inhibition of the DAF-16 translocation. It is possible that the DAF-16 targets are differentially expressed due to the translocation state of DAF-16 or involvement of other factors (e.g., other transcriptional regulators).

The upregulation of the longevity regulating pathway transcriptional targets generally contributes to increased lifespan. Since many of these targets were downregulated under simulated microgravity, we sought to identify the potential impact of their downregulation by acquiring their RNAi phenotype from WormBase [34]. Lifespan variants (WBRNAi00063155 and WBRNAi00063156) for *hsp-12.6* and extended lifespan (WBRNAi00064044) for *sod-3* have been reported. Follow-up experiments are needed to fully understand the role of these genes in the longevity regulation under simulated microgravity.

To further investigate the involvement of the longevity genes in simulated microgravity, we compared the DEGs to the DAF-16-responsive genes [51]. We found 144 DAF-16-induced genes were downregulated and this overlap was statistically significant (hypergeometric test, *p*-value < 0.0001). The number of shared genes between the simulated microgravity-induced genes and the genes induced or repressed by DAF-16 are 11 and 35, respectively (hypergeometric test, *p*-value > 0.05 for both). Similarly, the number of downregulated DAF-16-repressed genes were insignificant with a total of 57 shared genes (hypergeometric test, *p*-value > 0.05) (Appendix A). Together, our results suggest that the longevity regulation genes are affected under simulated microgravity and that along with the DAF-16-regulation, other mechanisms have an important function in this process.

### 3.4. Sphingolipid Signaling Pathway Is Suppressed in Response to Simulated Microgravity

Previous studies have suggested that the sphingolipid signaling pathway plays a role in the expression of DAF-16/FOXO-regulated genes [52,53]. In line with this, our results showed that the sphingolipid signaling pathway is downregulated under simulated microgravity. That is, putative glucosylceramidase 4 gene (*gba-4*) and putative sphingomyelin phosphodiesterase *asm-3* are downregulated and these downregulation patterns were sustained for eight days after return to ground conditions (Figure 4a). This downregulation pattern indicates an attenuation in the ceramide levels through a potential decrease in the degradation of sphingomyelin and glucosylceramide to ceramide. Along with the other biological functions such as autophagy, senescence, and apoptosis, the sphingolipid signaling pathway has critical functions in aging and longevity regulation [52,54,55]. The inhibition of this conserved pathway results in an extension of lifespan in animals from worms to humans [54,55]. For example, the inactivation of *asm-3* in *C. elegans* causes translocation of DAF-16 into the nucleus, promotion of DAF-16 target gene expression, and extension of lifespan by 14–19% [53].

To validate that the ceramide and sphingosine levels are indeed decreased in response to the aforementioned gene downregulation, we performed mass spectrometry analysis in ground control and simulated microgravity-exposed worms concurrently (Appendix A; Figure 4b,c). Our results revealed that total ceramide, total hexosylceramide (HexCer), and d18:1 sphingosine are lower (1.5-, 3.6-, and 1.4-fold, respectively) under simulated microgravity (*t*-test, *p*-value < 0.05). Similarly, sphingoid base (SB) levels decreased 1.9-fold under simulated microgravity, but this decrease was not significant (*t*-test, *p*-value = 0.055) (Figure 4b). Interestingly, the levels of total lactosylceramide (LacCer) increased 27-fold (*t*-test, *p*-value < 0.05). Increase in the LacCer and HexCer levels have been determined as biomarkers of aging in humans and murine [56] while LacCer C18:1 was unaffected during aging in *C. elegans* [54]. It is unclear why LacCer levels are elevated when the HexCer levels are decreased, but the overall pattern indicates a downregulation in the sphingolipid signaling pathway under simulated microgravity.

The levels of long acyl-chain ceramides (≥C24) are elevated with advancing age and in age-related diseases such as diabetes and cardiovascular disease while C20 and C26 ceramides are unaffected with aging or developmental stage in *C. elegans* [54,57,58]. To further decipher the potential effect of simulated microgravity on aging-related mechanisms, we quantified the levels of different acyl-chain ceramides (Appendix A; Figure 4c). We observed a decrease in d18:1-C24, and d18:1-C24:1 ceramides (1.2-, and 3.2-fold, respectively) under simulated microgravity (*t*-test, *p*-value < 0.05). It has been found that the inhibition of C24*–*C26 ceramide-inducing ceramide synthase HYL-1 causes improvements in neuromuscular function and age-dependent hypoxia and stress response [59,60]. Hence, the lower levels of C24 and C24:1 ceramides in the simulated microgravity-exposed worms may indicate their positive impact on the stress response and aging of the worms.

Along with their roles in aging and longevity, different acyl-chain ceramides have other distinctive functions. For example, C16-ceramide induces germ cell apoptosis [61] and C20*–*C22 ceramide functions in resistance to hypoxia [60]. We determined that ceramides with different acyl-chain showed altered levels in response to simulated microgravity. While d18:1-C16 and d18:1-C18 exhibited a decrease (2.8- and 1.6-fold, respectively), d18:1-C20 and d18:1-C22 ceramides exhibited an increase (3.7- and 1.5-fold, respectively) under simulated microgravity (*t*-test, *p*-value < 0.05). Collectively, our findings suggest that sphingolipid and insulin/IGF-1 pathways play a role in simulated microgravity response.

### 3.5. Identification of the Common Microgravity-Responsive Genes between Four-Day CERISE and Simulated Microgravity

To investigate the microgravity-responsive genes, we compared the results from our simulated microgravity experiment to those from four-day CERISE spaceflight experiment on *C. elegans* reported in NASA’s GeneLab database (https://genelab.nasa.gov/ (accessed on 1 January 2019)). For this analysis, we used GEO2R to determine the microarray DEGs (|log2 fold change (logFC)| > 2 and FDR < 0.05). We reasoned that the genes that are commonly differentially expressed in all the studies should be the core genes responding to microgravity.

We analyzed our RNA-seq data both with the Tuxedo pipeline [30] and DEseq2 [31] to confirm that the results are not dependent upon the analysis method. The results from both the methods were clustered together indicating their similarity (Figure 5b). The highest number of DEG overlap was between our experiment and four-day CERISE, and the overlap was significant for both the methods (hypergeometric test, *p*-value < 0.05) (Figure 5c). Since the overlap between the four-day CERISE and our results from the Tuxedo pipeline is higher, we decided to use the results from this pipeline throughout the manuscript. The results from DEseq2 are reported in the Appendix A (Appendix A; Appendix A).

We determined a total of 134 common DEGs between our experiment and four-day CERISE and categorized these 134 genes as the putative microgravity-responsive genes (Figure 5a,c; Appendix A). We discarded 16 DEGs showing conflicting patterns between the experiments (i.e., upregulated in one experiment and downregulated in the other) for further analyses. We reasoned that if the remaining genes are collaboratively involved in a biologically relevant process, they might have protein–protein interactions (PPIs) with each other. To test this hypothesis, we examined the enrichment for PPI of these genes by using the STRING database [39]. Among 118 DEGs, 58 had known PPIs with each other (FDR < 1.0 *×* 10^−16^) indicating their collaborative involvement in a biological process (Figure 5d). The GO analysis revealed that the microgravity-responsive genes play a functional role in locomotion, body morphogenesis, and collagen and cuticulin-based cuticle development (Figure 5e). Similarly, the products of microgravity-responsive genes exhibit enrichment for nematode cuticle collagen N-terminal and collagen triple helix repeat domains (Figure 5f). Together, our findings suggest that mainly the collagen genes are affected under microgravity and this effect is reproducible between the studies.

Next, we asked whether the human orthologs of the microgravity-responsive genes have relevant functions which might be affected in astronauts. We identified 64 human orthologs to 44 microgravity-responsive genes in worms (Appendix A). The human orthologs exhibited overrepresentation for protein domains including calpain large subunit domain III and calpain family cysteine protease (Appendix A). Given the reported upregulation of calpain under microgravity and its function in muscle atrophy [62,63], these results provide an additional support for the conserved microgravity-responsive genes in worms and humans.

Our analyses revealed that 20 putative TF genes are upregulated under simulated microgravity conditions and compared these to the four-day CERISE mission. Among the 20 upregulated putative TF genes, Y56A3A.28 and T24C4.2 showed upregulation during the four-day CERISE, indicating a potential transcriptional regulation role for these two in response to microgravity. Our results additionally suggested that the upregulation pattern of these TFs is maintained for eight and twelve days after the return, respectively. We could not find a human ortholog for T24C4.2 while the WormBase-suggested human ortholog of Y56A3A.28 is PRDM16. PRDM16 regulates the switch between skeletal muscle and brown fat cells by inhibiting the skeletal muscle development and gene expression and stimulating brown adipogenesis [64]. Therefore, Y56A3A.28 can be a strong candidate as a negative regulator of muscle development under microgravity conditions.

## 4. Discussion

In this study, we examined the simulated microgravity responsive gene expression patterns and their maintained levels for four, eight, and twelve days after return to ground conditions. Longevity regulating pathways such as insulin/IGF-1 and sphingolipid signaling were affected under simulated microgravity. We identified the putative microgravity-responsive genes by determining the common microgravity responsive transcriptomic signatures by incorporating our study to a previous one with the same exposure time. Moreover, we revealed that the microgravity-responsive genes are potentially conserved between worms and humans by performing function analysis for the human orthologs of the worm microgravity-responsive genes and identifying some of their potential transcriptional regulators.

Sphingolipid signaling pathway plays crucial biological roles such as lifespan regulation, apoptosis, and oxidative-stress response [52,54,65]. Our results revealed an overall downregulation of the sphingolipid pathway which is generally related to an increase in longevity [54]. A recent study identified decreased ASM-1 and ASM-2 levels in the ISS-housed worms indicating a similar downregulation in response to microgravity [66]. Mutant strains or RNAi knockdown of acid sphingomyelinase ASM-1, ASM-2, and ASM-3 genes increase the lifespan, ASM-3 being the most prominent one [53]. Overall, both simulated and the ISS-introduced microgravity seem to contribute to a potential increase in longevity through the downregulation of the sphingolipid signaling pathway. Follow-up experiments are needed to delimitate the contribution of sphingolipids on longevity under microgravity.

In the Twins Study of NASA, Scott Kelly was sent to the ISS for 340 days while his identical twin Mark Kelly remained on Earth. The results from this investigation showed that Scott Kelly’s telomere length significantly increased (14.5%) during the mission. In addition, the sphingolipid signaling pathway was differentially expressed in Scott Kelly. The telomere length of Mark Kelly, however, remained relatively stable [67]. Telomere length and lifespan have a strong relationship. That is, longer telomeres are linked to longer lifespan and increased resistance to environmental stress while shorter telomeres are linked to accelerated aging and reduced longevity [68,69,70,71]. Ceramide functions in the regulation of telomere length [55,72]. Considering our findings on decreased ceramide levels and the observations on increased telomere in the ISS, investigation of the ceramide and telomere length relationship might prove important to uncover the mechanisms affecting longevity in space.

Most of the space flight-triggered physiological changes are reverted after return to Earth [24,67]. Similarly, we observed that the majority of the simulated microgravity-induced DEGs (over 84%) are reverted 12 days after return to ground conditions. Our results have revealed that while some biological processes are only affected during the exposure, others can be affected for at least eight days (approximately two generations) after the return. For example, neuropeptide signaling pathway genes were differentially expressed only during the exposure whereas defense response genes remain downregulated for eight days after the return. Approximately 50% of the astronauts from Apollo mission experienced minor bacterial or viral infections at the first week of their return. Later studies reported that space flight-induced reactivation of latent herpes viruses which lasted for a week after the return to Earth (reviewed in [73]). These consistent observations indicate that the detrimental effects of space conditions (i.e., microgravity) on the immune system carry over even after the return. Given the longevity differences in worms and humans, future investigations could provide insight into whether the duration of the exposure has an impact on the duration of the lasting effects.

The limitation of our study is the inclusion of mixed stage worms and the collection of samples from whole animals. This limitation is common among the worm experiments conducted in the ISS [23,74,75]. Considering the significantly different gene expression patterns among developmental stages and across cell types [76], it is possible to retrieve biased results from the averaged gene expression levels. It is essential to address these issues in future studies.

The comparison of spaceflight responsive genes in *C. elegans* and *Drosophila melanogaster* demonstrated only six common genes from the European Soyuz flights to the ISS [17]. Similarly, the overlap between our results and four-day CERISE were relatively small. One of the reasons for these discrepancies can be the batch effects introduced due to the differences in the experimental designs. For instance, different exposure times to the space conditions or the usage of different food source or strain may induce distinct responses in the worms. It is crucial to examine the effects of individual factors (e.g., diet and gravitational force) along with the combinatory ones. Towards that end, we previously reported the impact of liquid cultivation of two *C. elegans* strains on gene expression and phenotype [26,77] and here, studied the impact of simulated microgravity.

## 5. Conclusions

Given the elevated interest in the human exploration of space, it is crucial to determine the detrimental effects of the space conditions on biological systems. We believe that our results will be a valuable reference for the studies on transcriptomic response to microgravity by controlling for the renowned batch effects from space conditions, allowing the worms to acclimate to liquid cultivation before the experiment, presenting the sustained transcriptomic responses after return to ground along with their putative regulators, integrating the results from a previous study conducted at the ISS, and providing the response of different ceramide profiles to simulated microgravity.

## Figures and Tables

**Figure 1 cells-12-00270-f001:**
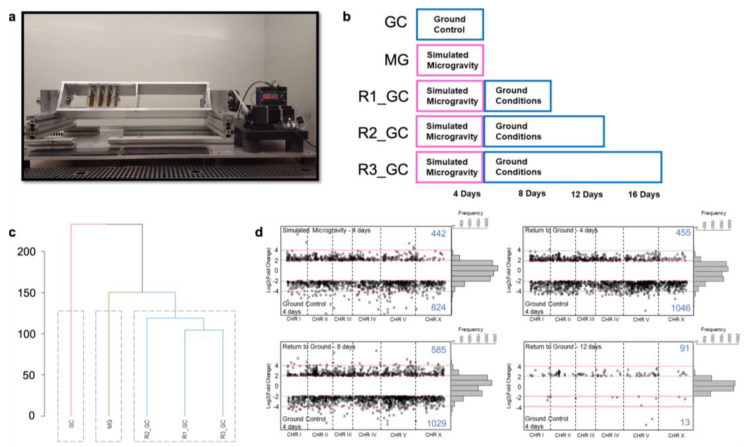
Transcriptomic response of *C. elegans* to simulated microgravity and return to ground conditions. (**a**) Clinostat used for simulating microgravity. (**b**) Experimental design for the effect of simulated microgravity on gene expressions during the exposure and after the return. (**c**) Dendrogram of the gene expression profiles for ground condition (GC), simulated microgravity (MG), four days after return to ground conditions (R1_GC), eight days after return to ground conditions (R2_GC), and twelve days after return to ground conditions (R3_GC). (**d**) Log2 fold change of the gene expressions (FPKM) between the conditions.

**Figure 2 cells-12-00270-f002:**
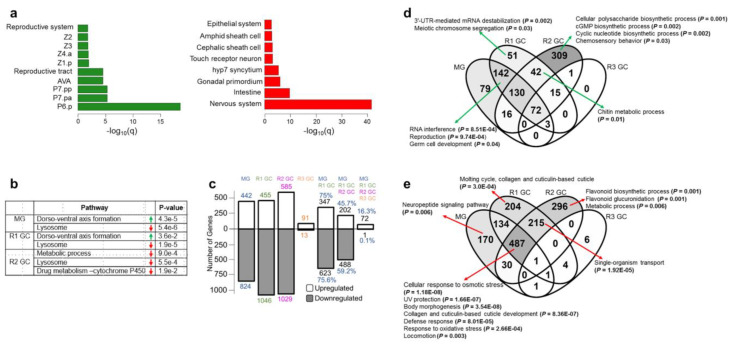
Differentially expressed gene profiles under simulated microgravity and after return to ground conditions. (**a**) Tissue enrichment of the upregulated (green) and downregulated (red) genes under simulated microgravity. (**b**) Pathway enrichment of the upregulated (green) and downregulated (red) genes during the exposure to simulated microgravity (MG), and four and eight days after return to ground conditions (R1 GC and R2 GC, respectively). (**c**) The number of differentially expressed genes under simulated microgravity and the transmission of the differential expression after return to ground conditions. (**d**) Categorization of the upregulated genes in comparison to the ground control animals, and the enriched gene ontology terms assigned to them. (**e**) Categorization of the downregulated genes in comparison to the ground control animals, and the enriched gene ontology terms assigned to them.

**Figure 3 cells-12-00270-f003:**
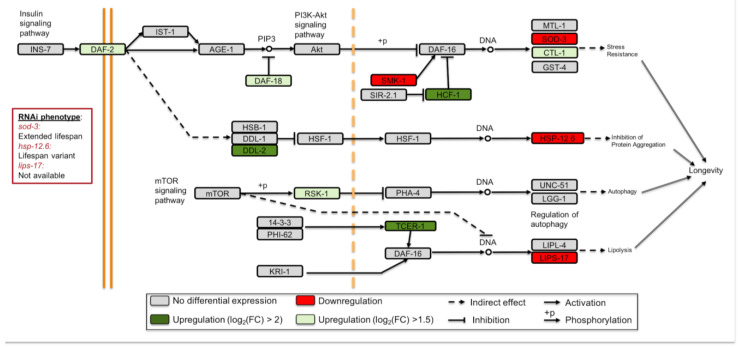
Longevity regulating pathway genes are differentially expressed under simulated microgravity. Adopted from the KEGG longevity regulating pathway—worm (cel04212).

**Figure 4 cells-12-00270-f004:**
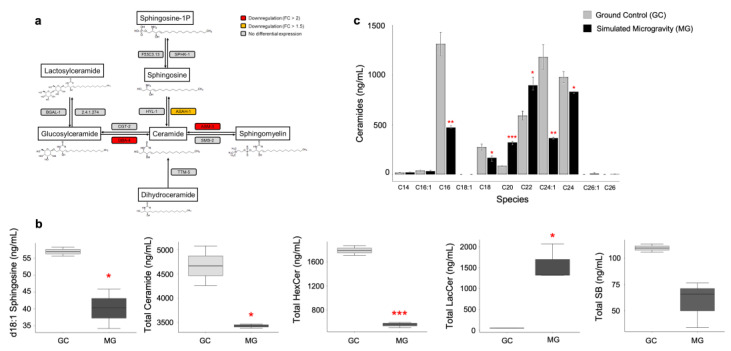
Sphingolipid signaling pathway is downregulated under simulated microgravity. (**a**) The sphingolipid signaling pathway genes *asah-1*, *asm-3*, and *gba-4* are downregulated under simulated microgravity. The downregulation pattern of *asm-3* and *gba-4* is maintained for eight days after return to ground conditions. (**b**) The levels of d18:1 sphingosine, total ceramide, hexosylceramide, and sphingoid base reduced while total lactosylceramide level increased under simulated microgravity. (**c**) The levels of different acyl-chain ceramides show alterations under simulated microgravity. The error bars represent the standard error of the mean (* *p* < 0.05, ** *p* < 0.01, *** *p* < 0.001).

**Figure 5 cells-12-00270-f005:**
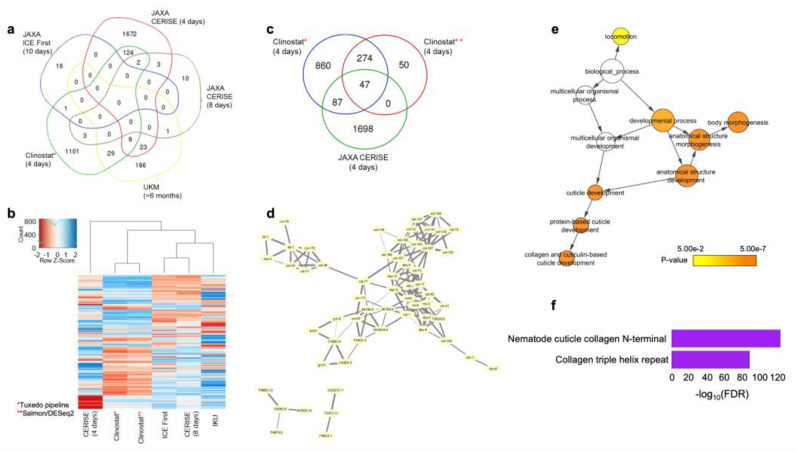
Microgravity-responsive genes are consistently differentially expressed in the ISS and under simulated microgravity. (**a**) Shared DEGs among the space-flown worms and our simulated microgravity experiment. The highest number of common DEGs are between our experiment and four-day CERISE. These genes are named as the “microgravity-responsive genes”. (**b**) The hierarchical clustering of the overall gene expressions among the space-flown worms and our simulated microgravity experiment. Our results analyzed with two different data analysis pipelines (*,**) demonstrated highly similar patterns. (**c**) The number of common DEGs were high for our results analyzed with two different pipelines (*,**). Both of the pipelines showed that the common number of DEGs are higher than the values expected by chance (hypergeometric text, *p* < 0.05). (**d**) Protein–protein interaction of the microgravity-responsive genes from STRING database. (**e**) Gene ontology enrichment of the microgravity-responsive genes. (**f**) Protein domain enrichment of the microgravity-responsive genes.

## Data Availability

All the sequencing data generated for this study have been submitted to the NCBI Gene Expression Omnibus (GEO; http://www.ncbi.nlm.nih.gov/geo/) under accession number GSE122125.

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
