# Peer review of "Transcriptomic Signature of the Simulated Microgravity Response in Caenorhabditis elegans and Comparison to Spaceflight Experiments"

_cells, 2023, doi:10.3390/cells12020270_

Round 1
Reviewer 1 Report
Here the authors decribed the effect of microgravity on gene expression using the small round worm C. elegans. While the experiment itself seems to be relative simple (4 days exposure, followed by gene expression analysis) in my opinion the authors described the experiments and the gene expression analysis very well and concise. Several comparisons with previous datasets were made and multiple different aspects have been taken into account. In the discussion their findings where nicely placed in the context of current knowledge and literature. I recommend publishing the manuscirpt.
I have two little comments: C. elegans gene names should be in italics and 2) I tried to open the link www.mdpi.com/xxxx/S1 as written down in the manuscript, but it doesn't work
Author Response
Comment 1: Here the authors described the effect of microgravity on gene expression using the small round worm C. elegans. While the experiment itself seems to be relative simple (4 days exposure, followed by gene expression analysis) in my opinion the authors described the experiments and the gene expression analysis very well and concise. Several comparisons with previous datasets were made and multiple different aspects have been taken into account. In the discussion their findings where nicely placed in the context of current knowledge and literature. I recommend publishing the manuscript.
Response 1: We thank the reviewer for the kind comments.
Comment 2: I have two little comments: C. elegans gene names should be in italics
Response 2: Gene names have now been converted to italic. We thank the reviewer for pointing this out.
Comment 3: and 2) I tried to open the link www.mdpi.com/xxxx/S1 as written down in the manuscript, but it doesn't work
Response 3: Since we currently do not have an www.mdpi.com link, we are unable to deposit the supplementary materials to the link provided with the template. However, all the supplementary files are submitted with the manuscript.
Reviewer 2 Report
In this article, the authors study the transcriptomic response of C. elegans to simulated microgravity and post-returning to ground conditions for four, eight, and twelve days. They characterize transcriptomic similarities and differences in these two conditions. Their analysis of genomics and metabolomics experiments demonstrates that simulated microgravity affects insulin/IGF-1 and sphingolipid signaling. However, the authors did not provide any evidence to show the involvement of these pathways in post-MG longevity. Can authors provide a simple survival assay analysis of N2 and mutants of a few candidate genes from these pathways and show no- and post-MG lifespans?
The authors should provide more details about RNA isolation and library preparation. In reference #26, total RNA from approximately 100,000 synchronized young adult C. elegans (4 h post-L4) were isolated with a modified TRIzol protocol and recovered by alcohol precipitation. However, at 22 degrees Celcius, worms take longer than 4 hours to reach the young adult stage. As having age synchronized population is critical for transcriptomics experiments, the authors should describe the exact growth conditions of worm harvesting for RNA isolation.
In the STRING database analysis, authors considered only interactions from experiments, gene fusion, databases, and co-expression. Authors should include genetic interactions or explain their exclusion.
Please follow the worm base gene nomenclature system for C. elegans genes.
Author Response
Comment 1: In this article, the authors study the transcriptomic response of C. elegans to simulated microgravity and post-returning to ground conditions for four, eight, and twelve days. They characterize transcriptomic similarities and differences in these two conditions. Their analysis of genomics and metabolomics experiments demonstrates that simulated microgravity affects insulin/IGF-1 and sphingolipid signaling. However, the authors did not provide any evidence to show the involvement of these pathways in post-MG longevity. Can authors provide a simple survival assay analysis of N2 and mutants of a few candidate genes from these pathways and show no- and post-MG lifespans?
Response 1: We thank the reviewer for this suggestion. However, currently we are unable to conduct these experiments as the grant award has been closed. We are in the process of submitting grant proposals to continue this study by targeting specific gene products based on these results and examining the associated phenotypes (including longevity assays).
Comment 2: The authors should provide more details about RNA isolation and library preparation. In reference #26, total RNA from approximately 100,000 synchronized young adult C. elegans (4 h post-L4) were isolated with a modified TRIzol protocol and recovered by alcohol precipitation. However, at 22 degrees Celsius, worms take longer than 4 hours to reach the young adult stage. As having age synchronized population is critical for transcriptomics experiments, the authors should describe the exact growth conditions of worm harvesting for RNA isolation.
Response 2: We apologize for not thoroughly proofreading the experimental summary, which as written is incorrect. We have included a detailed methods section, and revised the associated experimental schema in Figure 1.
Comment 3: In the STRING database analysis, authors considered only interactions from experiments, gene fusion, databases, and co-expression. Authors should include genetic interactions or explain their exclusion.
Response 3: We thank the reviewer for the suggestion. The STRING database currently reports interactions from the following: experiments, gene fusion, databases, co-expression, text-mining, fusion, neighborhood, and co-occurrence. Based on the descriptions in the STRING database, the neighborhood shows runs of genes that occur repeatedly in close neighborhood in (prokaryotic) genomes, the co-occurrence view shows the presence or absence of linked proteins across species, the fusion shows the individual gene fusion events per species, and text-mining demonstrates the (computationally) predicted interactions abstracts of scientific literature. Since we are only interested in PPIs with higher experimental and/or computational confidence and neighborhood, co-occurrence, fusion, and text-mining results are excluded from our analyses. We have now addressed this in page 3, lines 127-129.
Comment 4: Please follow the worm base gene nomenclature system for C. elegans genes.
Response 4: We are currently using Wormbase’s WBCel235 as a reference.
Round 2
Reviewer 2 Report
I am satisfied with the changes made by the authors and the relevant explanations for their inability to perform the suggested experimentations. These experiments do not challenge the claims/conclusions of the manuscript, and I support the publication of this manuscript. Comment 4: Please thoroughly check the manuscript for gene nomenclatures, e.g., C. elegans should be italic line#401Author Response
Comment 1: I am satisfied with the changes made by the authors and the relevant explanations for their inability to perform the suggested experimentations. These experiments do not challenge the claims/conclusions of the manuscript, and I support the publication of this manuscript.
Response 1: We thank the reviewer.
Comment 2: Comment 4: Please thoroughly check the manuscript for gene nomenclatures, e.g., C. elegans should be italic line#401
Response 2: Gene names in lines 323 and 345 have now been converted to italic. We thank the reviewer for pointing this out.